# TEMPORALLY-GROUNDED LANGUAGE GENERATION: A BENCHMARK FOR REAL-TIME VISION-LANGUAGE MODELS

## ABSTRACT

Vision-language models (VLMs) have shown remarkable progress in offline tasks such as image captioning and video question answering. However, real-time interactive environments impose new demands on VLMs, requiring them to generate utterances that are not only semantically accurate but also precisely timed. We identify two core capabilities necessary for such settings—*perceptual updating* and *contingency awareness*—and propose a new benchmark task, **Temporally-Grounded Language Generation (TGLG)**, to evaluate them. TGLG requires models to generate utterances in response to streaming video such that both content and timing align with dynamic visual input. To support this benchmark, we curate evaluation datasets from sports broadcasting and egocentric human interaction domains, and introduce a new metric, **TRACE**, to evaluate TGLG by jointly measuring semantic similarity and temporal alignment. Finally, we present **Vision-Language Model with Time-Synchronized Interleaving (VLM-TSI)**, a model that interleaves visual and linguistic tokens in a time-synchronized manner, enabling real-time language generation without relying on turn-based assumptions. Experimental results show that VLM-TSI significantly outperforms a strong baseline, yet overall performance remains modest—highlighting the difficulty of TGLG and motivating further research in real-time VLMs.

## 1 INTRODUCTION

With the success of large language models (LLMs) in producing fluent and contextually coherent text, turn-based chatbots have become one of the most widespread applications. This alignment between turn-based interaction and LLM optimization extends naturally to current vision-language models (VLMs), which pair vision encoders with pretrained LLMs to create multimodal chatbots capable of processing images or short video clips. While effective for tasks like image captioning and visual question answering, this paradigm breaks down in real-time or embodied environments, where inputs are continuous and responses must be generated on-the-fly without clear interaction boundaries.

Recent work (Bao et al., 2023; Chen et al., 2024) has identified the limitations of adapting turn-based VLMs to real-time settings, noting either high response latency or excessive computational overhead. A promising advance is VideoLLM-Online (Chen et al., 2024), which introduces a *streaming EOS prediction* task to allow VLMs to decide, frame-by-frame, whether to generate a response. However, VideoLLM-Online still assumes that the environment pauses during language generation–that is, it effectively assumes zero-latency responses–which in practice leads to delayed or overlapping utterances when deployed in real-time.

In this work, we focus on two key capabilities essential for real-time interactive VLMs: *perceptual updating*, the ability to revise ongoing interpretations based on new input, and *contingency awareness*, the ability to adjust actions based on their effects. To systematically evaluate these capabilities, we introduce **Temporally-Grounded Language Generation (TGLG)**, a benchmark task that requires models to generate utterances that are both semantically accurate and precisely timed in response to streaming visual input.

To support TGLG, we curate video-text datasets from two domains: play-by-play soccer broadcasts (SoccerNet (Cioppa et al., 2022)) to test perceptual updating, and egocentric human interactions (HoloAssist (Wang et al., 2023)) to test contingency awareness. We further introduce **Temporal Responsiveness and Alignment Coherence Evaluation (TRACE)**, a metric that jointly measures semantic relevance and temporal alignment between generated and ground-truth utterances.

Finally, we propose **Vision-Language Models with Time-Synchronized Interleaving (VLM-TSI)**, a new class of VLMs that align vision and text tokens along a shared timeline, enabling fluid, frame-by-frame generation without freezing observation. Our evaluations show that VLM-TSI outperforms the turn-based baseline, VideoLLM-Online, on TGLG under the TRACE metric, highlighting a promising direction for real-time VLM development.

Our contributions are fourfold: 1) we introduce the **TGLG** benchmark and the **TRACE** metric; 2) we curate datasets specifically targeting perceptual updating and contingency awareness; 3) we propose **VLM-TSI**, a real-time capable VLM architecture; and 4) we conduct extensive evaluations demonstrating VLM-TSI's advantages in real-time interaction scenarios.

## 2 RELATED WORK

### 2.1 VIDEO UNDERSTANDING BENCHMARKS

Early video understanding benchmarks (Chen & Dolan, 2011; Xu et al., 2016; Yu et al., 2019), inspired by the success of action recognition datasets (Soomro et al., 2012; Kay et al., 2017; Kuehne et al., 2011), focused on short video captioning. Following the success of LLMs and VLMs, a range of video question-answering benchmarks (Xu et al., 2017; Jang et al., 2017; Yu et al., 2019; Maaz et al., 2024) emerged.

The identification of "single-frame bias" (Lei et al., 2023), where a single frame suffices to answer questions, shifted attention toward long-form video understanding. EgoSchema (Mangalam et al., 2023) introduced "certificate length" to quantify how much video evidence is needed for verification, inspiring a wave of long-form benchmarks (Zhou et al., 2024a; Fu et al., 2024; Wang et al., 2024).

However, these benchmarks assume offline access to all frames. Recent work (Zhang et al., 2024; Zhou et al., 2024c; Xiong et al., 2025; Yang et al., 2025; Lin et al., 2024) targets streaming settings, where frames arrive sequentially. Motivated by these gaps, we propose a new benchmark that addresses real-time language generation under streaming video input.

### 2.2 VISION-LANGUAGE MODELS FOR STREAMING VIDEO

A growing body of research focuses on VLMs for streaming video, contrasting with earlier models for offline settings. Problem formulations vary widely: some emphasize real-time understanding (Lin et al., 2024; Zhang et al., 2024), while others prioritize embodied task execution (Bao et al., 2023; Wang et al., 2023).

Initial models targeted dense captioning (Zhou et al., 2024b) but lacked dialogue capabilities. Flash-VStream (Zhang et al., 2024) introduced dialogue over streaming input via a memory system but remains reactive. VideoLLM-Online (Chen et al., 2024) advanced toward proactive generation, dynamically responding to evolving content.

We extend this proactive direction by proposing a time-synchronized interleaving strategy and a benchmark focused on evaluating perceptual updating and contingency awareness.

## 3 LIMITATIONS OF TURN-BASED VLMS IN REAL-TIME ENVIRONMENTS

VLMs owe most of their powerful capabilities, such as visual reasoning and understanding, to their LLM backbones. As a result, they inherit a fundamental assumption from LLMs: that interactions are turn-based, where the environment pauses while the model generates responses (i.e., zero-latency language generation) and vice versa. Unfortunately, this assumption introduces significant latency and coherence issues in real-time embodied environments, where turns are not clearly defined and inputs arrive continuously.

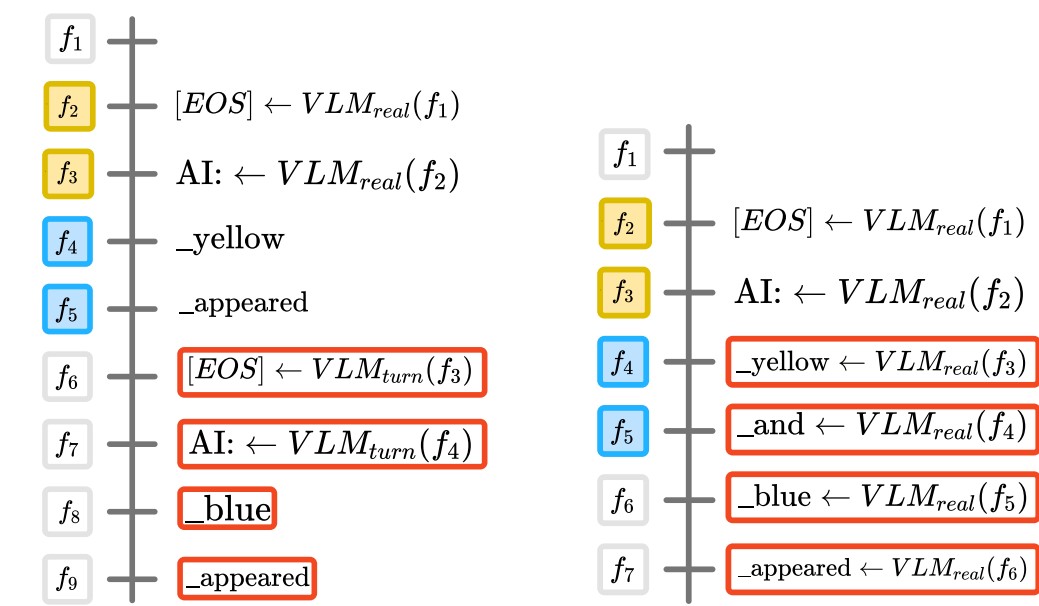

(a) In practice, frames continue to stream while the model generates. As a result, the turn-based model's utterance about the yellow frame's appearance delays the processing of the blue frame. Red boxes indicate utterances that are generated out of sync with the visual context.

(b) A real-time model begins responding when the yellow frame appears and detects the blue frame mid-generation. It updates the utterance (red box) to reflect this new perceptual evidence, maintaining temporal alignment with the environment.

Figure 1: Turn-based VLMs fail to operate effectively in real-time environments, because they cannot process new perceptual input while generating responses. $[EOS]$ denotes no generation.

To illustrate, consider a simple scenario in which a VLM is tasked with notifying a user when a colored frame appears. Assume that the environment is represented as a stream of video frames, with each frame arriving at the same rate the model can generate a single token. A turn-based VLM such as VideoLLM-Online erroneously assumes that once it begins generating an utterance (e.g., to notify the user that a yellow frame has appeared), the environment effectively pauses and no new frames are streamed. In practice, video frames continue to arrive while the model generates its response. This results in a mismatch: the model's utterance about the yellow frame's appearance delays recognizing and responding to the blue frame (Figure 1a). These misalignments compound as the interaction continues, leading to increasingly out-of-sync and less useful responses.

While this may seem like a simple issue of timing, it reveals a deeper limitation: the model cannot revise or adapt its output in response to new input that arrives mid-generation. This deficiency points to the absence of two core capabilities identified in cognitive psychology as essential for real-time interaction. The first is *perceptual updating*—the ability to continuously integrate new sensory input and revise ongoing interpretations accordingly. The second is *contingency awareness*—the ability to understand how one's actions influence the environment and adjust behavior in response. A turn-based VLM, by design, cannot update its output on the fly or respond to how its utterances impact the environment, making it fundamentally misaligned with the demands of real-time, interactive environments.

In contrast, a real-time VLM would begin generating an utterance when the yellow frame appears, detect the blue frame mid-generation, and revise the ongoing utterance accordingly to reflect this change (Figure 1b). This behavior highlights the real-time adaptation needed to enable both perceptual updating and contingency awareness—capabilities fundamentally at odds with turn-based assumptions. This motivates a shift toward models that continuously couple perception and generation, allowing new input to inform and revise ongoing output in real time.

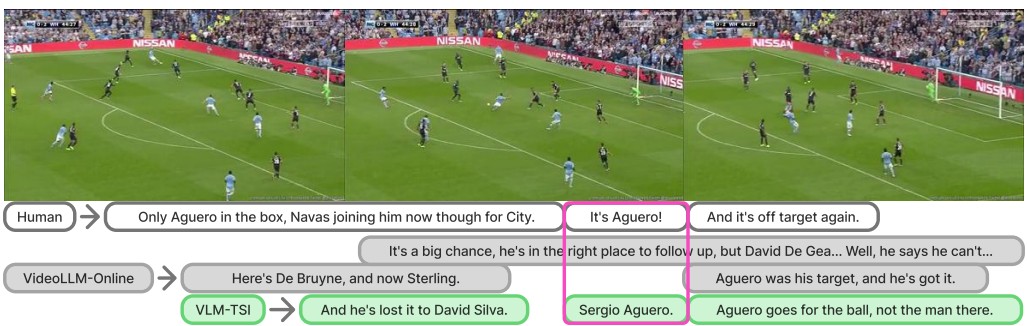

Figure 2: Sports broadcast datasets like SoccerNet (Cioppa et al., 2022) contain dynamic visual events that require robust perceptual updating. Turn-based models like VideoLLM-Online produce semantically and temporally inaccurate utterances with unrealistic overlaps, while real-time models like VLM-TSI generate semantically aligned and precisely timed utterances without overlaps.

## 4 TEMPORALLY-GROUNDED LANGUAGE GENERATION

To our knowledge, there are currently no benchmarks that jointly evaluate perceptual updating and contingency awareness in VLMs operating under real-time constraints. As discussed in Section 3, generating utterances that are both semantically meaningful and temporally aligned is essential for supporting these two capabilities. While "on-policy" evaluation would be ideal, building such a protocol is tantamount to solving real-time interactive environments themselves. In contrast, a well-designed "off-policy" evaluation is far easier to construct and use, enabling faster iteration and model development. We formalize these requirements in a new task, which we call **Temporally-Grounded Language Generation (TGLG)**, to facilitate the development and evaluation of VLMs in real-time environments.

### 4.1 DATA CURATION

Perceptual updating and contingency awareness are fundamentally tied to the ability to generate utterances that are not only semantically appropriate but also precisely timed. To ensure that our benchmark meaningfully tests these capabilities, we carefully curate evaluation datasets in which both the content and timing of model responses are critical.

#### 4.1.1 PERCEPTUAL UPDATING

To evaluate a model's capacity for perceptual updating, we seek interactions where the visual scene changes rapidly and continuously, sometimes even within a single utterance. Sports broadcasting provides a natural setting for this: commentators, especially "play-by-play" commentators (who narrate unfolding events) rather than "color" commentators (who provide analysis and background), must respond quickly to unfolding gameplay, often revising or elaborating on their observations as new events occur in real time.

We use the SoccerNet dataset (Cioppa et al., 2022) as our source of sports broadcasting videos with live commentary audio (Figure 2). The audio is transcribed with WhisperX (Bain et al., 2023), and we filter out clips with no speech, non-English speech, or non–play-by-play segments such as analysis, banter, and advertisements (see Section A for details). From the remaining play-by-play commentary, we extract evaluation interaction histories as defined in Section 4.2, ensuring that each segment exhibits tight temporal coupling between video events and commentary.

The final dataset is split into training and test sets in an 80/20 ratio, with the training set further divided into training and validation splits using the same proportion.

#### 4.1.2 CONTINGENCY AWARENESS

To evaluate contingency awareness, we seek interactions where the model's utterances directly influence the visual scene. Egocentric human interaction datasets, often used for the development of

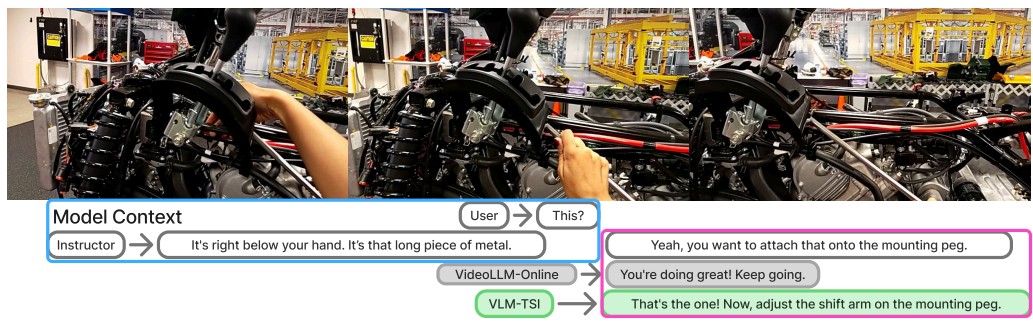

Figure 3: Egocentric interaction datasets like HoloAssist (Wang et al., 2023) capture complex cooperative interactions that require robust contingency awareness. Turn-based models like VideoLLM-Online may produce temporally aligned utterances, but they struggle to generate useful instructions because they fail to account for the consequences of their prior outputs. In contrast, real-time models like VLM-TSI reason over their past utterances and adapt to the evolving scene, resulting in more context-aware guidance.

task-guidance systems, naturally support this property. These datasets typically involve a human user performing a task while wearing an egocentric camera, guided by a human instructor who issues instructions in response to the live egocentric video feed of the user. As a result, the instructor's utterances directly shape the user's actions, which in turn alter the egocentric visual input.

We use the HoloAssist dataset (Wang et al., 2023) as our source of egocentric human interaction videos (Figure 3). HoloAssist includes transcribed dialogues annotated with fine-grained dialogue acts, which we use to identify moments that require contingency awareness. Specifically, we extract interactions that begin with an instruction from the instructor (dialogue act: `instructor-start-conversation_describing high-level instruction`) and end with a correction (dialogue act: `instructor-start-conversation_correct the wrong action`). These segments capture scenarios where the instructor's initial utterance prompts an action, a mistake is observed, and a corrective instruction follows, resulting in visual scene changes that depend on earlier model outputs.

Due to the limited number of such interactions, we use the entire curated subset as a held-out test set.

Detailed dataset statistics are provided in Section B. Note that while our benchmark is curated from two domains, it targets the general capabilities of perceptual updating and contingency awareness rather than domain-specific knowledge, and thus we expect the evaluation results to generalize well beyond these settings.

## 4.2 Task Definition

We define TGLG (Temporally-Grounded Language Generation) using timestamped utterances from curated datasets. To illustrate the task, consider the following example from HoloAssist:

1. 33.2-43.3: "Assistant: Now remove the indicated component that's damaged, . . ."
2. 45.3-46.6: "User: Oh, this thing?"
3. 46.6-47.4: "Assistant: To the right."
4. 47.9-49.2: "Assistant: The small cube."
5. 49.3-49.8: "Assistant: Yes."

This segment illustrates both perceptual updating and contingency awareness: the assistant must detect and correct the user's misunderstanding (utterances 3–4) and confirm the correct action (utterance 5). Concretely, we provide the model with utterances 1 and 2 along with their associated video frames, then stream the remaining frames. The model must generate grounded, time-sensitive responses based on the streamed frames. If it exhibits perceptual updating and contingency awareness, its generated utterances should closely match the human references (utterances 3-5) in both timing and content. Note that utterance 1 would be used as the human reference for the previous evaluation instance. For models that do not emit explicit end timestamps, we estimate utterance

durations based on token count and a fixed speech rate. Further details and formal definitions are provided in Appendix C.

## 4.3 METRIC

To evaluate perceptual updating and contingency awareness in real-time settings, we introduce **Temporal Responsiveness and Alignment Evaluation (TRACE)**, a comprehensive metric for the TGLG benchmark.

TRACE jointly measures *semantic accuracy* and *timing precision* by aligning generated and ground-truth utterances based on temporal proximity. The final score is a weighted combination:

$$\text{TRACE} = \alpha S^a + (1 - \alpha)S^t \tag{1}$$

where $S^a$ captures semantic accuracy and $S^t$ reflects timing alignment. The timing score $S^t$ is further decomposed into:

$$S^t = \alpha_{\text{start}}S^{\text{start}} + \alpha_{\text{end}}S^{\text{end}} + (1 - \alpha_{\text{start}} - \alpha_{\text{end}})S^{\text{overlap}} \tag{2}$$

where $S^{\text{start}}$ and $S^{\text{end}}$ measure start and end time alignment, and $S^{\text{overlap}}$ penalizes overlapping utterances. All components are scaled by an F1-based alignment score to penalize over- and under-generation.

By jointly evaluating *what* is said and *when* it is said, TRACE captures the dual demands of real-time interaction: adapting to new observations (perceptual updating) and responding to the consequences of prior actions (contingency awareness). It also promotes natural, human-like generation by evaluating model outputs against gold-standard human utterances. Because it focuses on these fundamental capabilities rather than domain-specific knowledge, TRACE supports evaluation that generalizes beyond sports broadcasting and egocentric interaction. Full details of calculating TRACE are provided in Appendix D.

## 5 VISION-LANGUAGE MODEL WITH TIME-SYNCHRONIZED INTERLEAVING

Current VLMs, such as VideoLLM-Online, assume turn-based interactions where the environment is effectively paused while the model generates a full utterance (i.e., zero-latency language generation) and vice versa. This design causes them to struggle in real-time settings where perceptual updating and contingency awareness are critical. To address this limitation, we introduce a new class of VLMs, called **Vision-Language Models with Time-Synchronized Interleaving (VLM-TSI)**, which drop the turn-based assumption and serve as a baseline for the TGLG task.

### 5.1 TIME-SYNCHRONIZED INTERLEAVING

The core idea behind VLM-TSI is that language generation and visual observation should proceed along a shared timeline, rather than in alternating turns. Unlike conventional VLMs that generate complete utterances before resuming observation, VLM-TSI alternates between ingesting new video frames and generating text tokens in a **temporally synchronized** manner. Specifically, it interleaves vision tokens $v_t$ (produced by the vision encoder) and text tokens $x_\tau$ into a single sequence ordered by timestamp such that each $x_\tau$ is conditioned on all visual and linguistic context observed up to that point (Figure 4).

### 5.2 TRAINING

VLM-TSI is trained using standard causal language modeling, with losses computed *only* for text tokens (using right-shifted labels). This is a departure from the "streaming EOS prediction" task used by VideoLLM-Online, which uses the EOS token as the label for vision tokens to signify silence. Chen et al. (2024) have observed that the streaming EOS prediction task tends to bias the model toward silence due to label imbalance, and they mitigate this by introducing a probability threshold below which EOS is not emitted. VLM-TSI instead focuses solely on learning when to generate text by predicting the BOS token following visual input. Our results empirically show that this strategy resolves the label imbalance without requiring threshold-based heuristics.

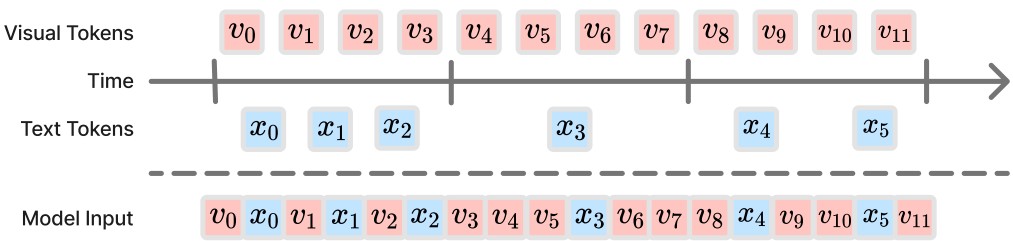

Figure 4: VLM-TSI interleaves vision tokens $v_t$ and text tokens $x_\tau$ in a temporally synchronized manner. For simplicity, each frame $f_t$ is encoded as a single vision token $v_t$.

### 5.3 INFERENCE

At inference time, VLM-TSI receives one visual token per timestep (or more if video frames are sampled faster than text generation) and performs one decoding step. If the model predicts a token other than BOS (i.e., it is not the start of a new utterance), it is discarded and the model waits for the next visual input. On the other hand, if the predicted token *is* a BOS token, the model enters *text generation mode* and continues decoding until the next vision token arrives, or an EOS or BOS token is generated, signaling utterance completion or new utterance start.

All generated text tokens are added to the context while BOS and EOS tokens are not. This process allows VLM-TSI to start and stop speaking dynamically in response to incoming visual observations without freezing the timeline.

## 6 EVALUATION

We now present the evaluation results of two baseline VLMs on the TGLG benchmark: VideoLLM-Online (Chen et al., 2024) and our proposed VLM-TSI.

### 6.1 EXPERIMENTAL SETUP

We use VideoLLM-Online as the primary baseline for TGLG and introduce VLM-TSI as a strong alternative that natively supports TGLG. We do not include conventional offline VLMs as additional baselines, as they are ill-suited for real-time settings: their per-frame autoregressive decoding over entire utterances incurs high computational costs, making them impractical for streaming applications where tokens must be generated frame-by-frame.

To ensure a fair comparison, we closely follow the training recipe from VideoLLM-Online Chen et al. (2024), modifying only the token interleaving strategy during fine-tuning. All models are trained with a video frame sampling rate of 2 FPS and use LoRA (Hu et al., 2022) with rank 128 and scaling factor 256 applied to all linear layers.

For perceptual updating, both models are initialized from the pretrained `VideoLLM-online-8B-v1+` checkpoint and fine-tuned on our curated SoccerNet training split (Section 4.1) for 5 epochs, which takes about 2 hours on four 48GB L40S nodes. For contingency awareness, we fine-tune VLM-TSI on the Ego4D Goal-Step streaming narration and dialogue data (Song et al., 2023; Chen et al., 2024), which is similar in domain to HoloAssist, for 2 epochs. This takes approximately 19 hours on two 48GB A40 nodes. Since VideoLLM-Online is already pre-trained on this dataset, we do not fine-tune it further. Furthermore, both models receive the summary of each activity, provided in HoloAssist, as part of the system message to provide high-level task context.

We set the EOS prediction threshold for VideoLLM-Online to the default value of 0.725 Chen et al. (2024) for perceptual updating. For contingency awareness, we increase this threshold to 0.8, as the model otherwise generated too few utterances.

For all evaluations, we set $\alpha_{\text{start}} = 0.4$, $\alpha_{\text{end}} = 0.4$, and $\alpha = 0.5$. For the sentence embedder emb($\cdot$), we use `all-mpnet-base-v2` from SentenceTransformers (Reimers & Gurevych, 2019).

Table 1: TGLG evaluation results. Best scores in bold.

| Capability | Model | TRACE | $S^a$ | $S^t$ | $S^{start}$ | $S^{end}$ | $S^{overlap}$ | $F_1$ |
|---|---|---|---|---|---|---|---|---|
| Perceptual Updating | VLM-TSI | **39.1** | **45.7** | **32.4** | **29.1** | **15.6** | **72.8** | **72.8** |
| | VideoLLM-Online | 27.1 | 32.4 | 21.8 | 24.4 | 11.1 | 37.9 | 51.0 |
| Contingency Awareness | VLM-TSI | **18.8** | **23.0** | **14.5** | **11.9** | **6.0** | **36.7** | **36.7** |
| | VideoLLM-Online | 9.6 | 12.3 | 6.8 | 5.9 | 4.0 | 14.4 | 18.2 |

## 6.2 RESULTS

In this section, we first present the overall evaluation results on both capabilities, followed by detailed analyses for each dataset.

### 6.2.1 OVERALL RESULTS

Table 1 summarizes overall TGLG evaluation results for VLM-TSI and VideoLLM-Online. VLM-TSI outperforms VideoLLM-Online across both capabilities and all sub-metrics, demonstrating more effective real-time behavior.

The most substantial gains come from the overlap score ($S^{overlap}$), where VLM-TSI nearly doubles the performance of VideoLLM-Online. This reflects a key architectural strength: VLM-TSI interleaves vision and language along a shared timeline, enforcing non-overlapping utterance generation by design. In contrast, turn-based models like VideoLLM-Online often produce overlapping utterances due to their delayed decoding behavior.

VLM-TSI also achieves better alignment in $S^{start}$ and $S^{end}$, though the relatively low absolute scores, especially for $S^{end}$, highlight a persistent challenge: models are better at detecting when to begin speaking than when to stop. Accurately terminating generation based on continuously evolving perceptual input remains difficult, particularly under tight latency constraints.

Both models perform worse on contingency awareness than on perceptual updating, consistent with the intuition that passively describing visual input (as in SoccerNet) is easier than generating language that influences downstream user actions (as in HoloAssist). This gap reinforces the difficulty of reasoning about causality and long-term consequences in real time.

Notably, even VLM-TSI achieves only moderate TRACE scores (e.g., 39.1 for perceptual updating, 18.8 for contingency awareness), underscoring that TGLG remains a challenging benchmark. The TRACE metric helps clarify these limitations by separately evaluating semantic and temporal alignment. Qualitative examples of common failure modes, such as delayed generation, premature cutoffs, and overlapping utterances, are included in Appendix G.

As for computational efficiency–an important consideration for real-time settings–VLM-TSI shares the same architecture as VideoLLM-Online aside from input token structuring, and thus inherits similar runtime characteristics. On an NVIDIA RTX 4090, we observe a throughput of 11.4 FPS without using the FIFO queue employed by VideoLLM-Online to parallelize video frame encoding with LLM decoding. This is comparable to reported VideoLLM-Online performance (10–15 FPS on an NVIDIA A100 and 5–10 FPS on a NVIDIA RTX 3090 (Chen et al., 2024)).

### 6.2.2 SOCCERNET RESULTS

We conduct further analysis on SoccerNet (perceptual updating). Using annotations from the SoccerNet Action Spotting Challenge Deliege et al. (2021), we first align each utterance with nearby game events (~5 seconds from start and end times). We then group the events into six high-level categories (Section E) and compute the difference in TRACE scores between the two models within each group.

As shown in Table 2, VLM-TSI outperforms VideoLLM-Online across all action categories, demonstrating its robust perceptual updating capabilities. The performance gap is narrowest for the "Goal/Penalty" category. We hypothesize that this is because such events are rare and temporally atomic, typically covered with a single utterance, so the delayed decoding pattern of turn-based models like VideoLLM-Online does not lead to significant performance degradation.

Table 2: Per-action comparison on SoccerNet: Difference in TRACE scores between VLM-TSI and VideoLLM-Online. Positive values indicate VLM-TSI outperforms VideoLLM-Online.

| Action Group | # actions | $\Delta$TRACE | $\Delta S^a$ | $\Delta S^t$ |
|---|---|---|---|---|
| Attempts | 3 | .12 | .12 | .12 |
| Discipline | 3 | .10 | .09 | .10 |
| Goal/Penalty | 2 | .06 | .09 | .03 |
| Infractions | 2 | .11 | .12 | .11 |
| Restarts | 6 | .14 | .14 | .13 |
| Substitution | 1 | .12 | .15 | .08 |

Table 3: Per-task comparison on HoloAssist: Difference in TRACE scores between VLM-TSI and VideoLLM-Online. Positive values indicate VLM-TSI outperforms VideoLLM-Online.

| Task Group | # tasks | $\Delta$TRACE | $\Delta S^a$ | $\Delta S^t$ |
|---|---|---|---|---|
| Assemble Furniture | 4 | .13 | .16 | .10 |
| Disassemble Furniture | 4 | .14 | .16 | .12 |
| Make Coffee | 2 | .11 | .13 | .08 |
| Repair Machinery | 3 | -.01 | -.02 | -.01 |
| Setup Electronics | 7 | .09 | .11 | .08 |

Notably, VideoLLM-Online did not generate any utterances aligned with the "Yellow→Red card" action from the "Infractions" category. We believe this highlights a key advantage of VLM-TSI's time-synchronized interleaving strategy. These actions consist of two rapid sub-events—a yellow card immediately followed by a red card—and are difficult to describe coherently in a single turn. VideoLLM-Online struggles to capture this sequence due to its turn-based decoding: once it begins generating, it cannot adjust its generation based on new visual input. In contrast, VLM-TSI can begin generating an utterance in response to the yellow card and seamlessly adapt mid-generation when the red card is shown, resulting in more accurate and temporally aligned output.

### 6.2.3 HoloAssist Results

To further analyze model performance on HoloAssist (contingency awareness), we group the tasks into five categories (Section F) and compare TRACE scores between the two models within each group.

As shown in Table 3, VLM-TSI outperforms VideoLLM-Online across all task categories except "Repair Machinery," where VideoLLM-Online slightly outperforms. We hypothesize that this exception stems from the nature of these tasks: repair sequences typically involve long, well-delimited, and visually salient steps. In such cases, the delayed decoding of turn-based models like VideoLLM-Online is less problematic.

In contrast, tasks such as "Assemble/Disassemble Furniture" often involve small, hard-to-distinguish physical manipulations, while "Make Coffee" and "Setup Electronics" frequently require interacting with appliances that display status updates or prompts on small screens. In these scenarios, VLM-TSI's ability to incorporate visual input incrementally and adjust utterances mid-generation provides a significant advantage, helping it better track and describe nuanced visual cues in real time.

## 7 Conclusion

In this work, we introduce **TGLG**, a benchmark task for evaluating two core capabilities essential to real-time VLMs: *perceptual updating* and *contingency awareness*. Alongside curated video–text datasets from sports broadcasts and egocentric interactions, we propose **TRACE**, a metric that jointly evaluates semantic relevance and timing precision, and **VLM-TSI**, a baseline that interleaves vision and text tokens along a shared timeline. Together, these contributions establish a foundation for building VLMs that are accurate, responsive, temporally adaptive, and capable of seamless real-time interaction.

REPRODUCIBILITY STATEMENT

Full task definitions, as well as the details of dataset curation and metric definitions, are provided in Sections 4. Full model definitions are provided in Section 5. The curated datasets for TGLG and the code for baseline training and evaluation will be fully released upon acceptance.

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

## A  NON-PLAY-BY-PLAY COMMENTARY FILTERING

Play-by-play announcers are trained professionals whose role is to narrate matches in real time with high semantic and temporal precision. Their utterances are typically short, declarative, and event-driven, often featuring player or team names and concise descriptions of unfolding actions. For example, "Swansea corner," "Routledge goes down inside the box," or "Good pressing again from Manchester United" illustrate this temporally grounded style. Such commentary is readily distinguished from analysis, banter, or advertisements based on text alone, without requiring frame-level review.

To automate filtering, we first manually annotated utterances from five matches with binary labels (play-by-play vs. other), then split them into training, validation, and test sets (70/15/15). A lightweight LSTM classifier trained on this data achieved 94% accuracy on the test set, identifying 58,031 play-by-play utterances out of 148,533 total (39%). Alignment between commentary and video content was further verified during development by inspecting paired examples.

## B  DATASET STATISTICS

Table 4: Dataset statistics for perceptual updating (SoccerNet Cioppa et al. (2022)) and contingency awareness (HoloAssist Wang et al. (2023)) benchmarks. **Size** indicates the number of datapoints. **Avg. Utt.** is the average number of utterances per datapoint. **Avg. Len.** is the average number of tokens per utterance. **Avg. Gap** is the average time in seconds between successive utterances within each datapoint.

| Capability | Source | Size | Avg. Utter. | Avg. Len. (tokens) | Avg. Gap (s) |
|---|---|---|---|---|---|
| Perceptual Updating | SoccerNet | 16487 | 5.67 | 10.98 | 1.13 |
| Contingency Awareness | HoloAssist | 1761 | 15.84 | 13.69 | 7.94 |

## C  TASK DEFINITION

In this section, we provide the full details of our proposed **Temporally-Grounded Language Generation (TGLG)** task. We consider the same example from HoloAssist presented in Section 4.2:

1. 33.2-43.3: "Assistant: Now remove the indicated component that's damaged, . . ."

2. 45.3-46.6: "User: Oh, this thing?"
3. 46.6-47.4: "Assistant: To the right."
4. 47.9-49.2: "Assistant: The small cube."
5. 49.3-49.8: "Assistant: Yes."

Each utterance $u_i = (s_i, e_i, \{x_t \mid s_i \leq t \leq e_i\})$ includes a start time $s_i$, an end time $e_i$, and a sequence of text tokens $x_t$ timestamped at $t$. The complete set of utterances in a video forms the interaction history $\mathcal{U} = \{u_i \mid 1 \leq i \leq N\}$.

We define *evaluation clusters* $\{u_i \mid i \in \mathcal{E}_j\}$ from this history. In HoloAssist, $\mathcal{E}_j$ contains instructor utterances that test contingency awareness (e.g., $\{u_3, u_4, u_5\}$ in the example); in SoccerNet, it includes play-by-play commentary. Each cluster includes utterances whose start and end timestamps fall within a 5-second window.

The *evaluation interaction history* is:
$$\mathcal{H}_j = \{u_i \mid 1 \leq i < \min(\mathcal{E}_j)\} \cup \{u_i \mid i \in \mathcal{E}_j\} \tag{3}$$
which includes all prior utterances as context and the cluster utterances as targets (e.g., $\{u_1, u_2\} \cup \{u_3, u_4, u_5\}$ in the example).

The model's input context is:
$$C_j = \{f_t \mid s_1 \leq t < s_{\min(\mathcal{E}_j)}\} \cup \{x_\tau \mid s_1 \leq \tau < s_{\min(\mathcal{E}_j)}\} \tag{4}$$
where $f_t$ is the video frame at time $t$, and $x_\tau$ is any observed token (system messages, prior dialogue, etc.) at time $\tau$. The video and text streams may have different sampling rates and are not assumed to be aligned. In the example, $C_j$ includes all frames up to 46.6 seconds and utterances $\{u_1, u_2\}$.

The model is evaluated on its outputs aligned to frames in:
$$\{f_t \mid s_{\min(\mathcal{E}_j)} \leq t \leq e_{\max(\mathcal{E}_j)}\} \tag{5}$$
and must produce utterances that are semantically appropriate and temporally aligned with the ground-truth utterances in $\mathcal{E}_j$ (e.g., $\{u_3, u_4, u_5\}$ in the example).

Because most turn-based VLMs emit utterances without end times, we estimate durations by assuming a speech rate of 150 words per minute [1] and 1.3 tokens per word [2] to infer the end time from the number of generated tokens.

# D    METRIC

In this section, we provide the full details of our proposed evaluation metric for TGLG, **Temporal Responsiveness and Alignment Evaluation (TRACE)**.

## D.1    ALIGNING GENERATED AND GROUND-TRUTH UTTERANCES

We begin by aligning the ground-truth utterances $\mathcal{U} = \{u_i \mid 1 \leq i \leq N\}$ (defined in Section 4.2) and the generated utterances $\hat{\mathcal{U}} = \{\hat{u}_j \mid 1 \leq j \leq M\}$ through bi-partite matching based on temporal proximity. We define a cost matrix $A \in \mathbb{R}^{N \times M}$ as:
$$A_{ij} = -\exp\left(-\frac{|s_i - \hat{s}_j|}{\tau_{\text{time}}}\right) \tag{6}$$
where $s_i$ and $\hat{s}_j$ are the start times of the ground-truth and generated utterances, respectively, and $\tau_{\text{time}}$ is a time-scale parameter that downweights matches between utterances with large start-time differences. This yields an initial one-to-one alignment based purely on temporal structure. Note that some matched pairs may be pruned based on timing thresholds.

To avoid penalizing semantically accurate utterances that are generated slightly out of order, we refine these matches via local optimization. Specifically, we compute a similarity matrix $\mathcal{S} \in \mathbb{R}^{N \times M}$ using cosine similarity between sentence embeddings:
$$\mathcal{S}_{ij} = \frac{1 + \cos\left(\text{emb}(u_i), \text{emb}(\hat{u}_j)\right)}{2} \tag{7}$$

---

[1]Source: Baruch College Tools for Clear Speech
[2]Source: OpenAI token documentation

where $\mathrm{emb}(\cdot)$ denotes a pretrained sentence embedding model, and cosine similarities are rescaled to lie in $[0, 1]$. We then iteratively refine the alignment by greedily swapping matched pairs $(i, j)$ and $(i', j')$ if:

$$\mathcal{S}_{ij} + \mathcal{S}_{i'j'} < \mathcal{S}_{ij'} + \mathcal{S}_{i'j} \tag{8}$$

and all four utterances involved fall within $\tau_{\mathrm{win}}$ of one another. This process is repeated for a fixed number of passes or until convergence.

Finally, we discard any matched pair $(i, j)$ for which $|s_i - \hat{s}_j| > \tau_{\mathrm{win}}$, ensuring temporal plausibility in the final alignment:

$$B = \{(i, j) \mid u_i \in \mathcal{U}, \hat{u}_j \in \hat{\mathcal{U}}, (i, j) \text{ is a matched pair}\} \tag{9}$$

For all evaluations, we set $\tau_{\mathrm{time}} = 3.0$ and $\tau_{\mathrm{win}} = 5.0$.

### D.2 SEMANTIC ACCURACY SCORE

The semantic accuracy score is computed over the set of matched pairs $B$ between generated and ground-truth utterances. While this score can be derived from human evaluation or LLM-based evaluation (Maaz et al., 2024), we adopt a semantic similarity-based approach (Yu et al., 2024) for its efficiency and reproducibility. Specifically, we use the similarity matrix $\mathcal{S}$ defined in Equation 7 to calculate the mean similarity over all matched utterance pairs and scale it by the generation $F_1$ score to penalize over- or under-generation:

$$S^a = \frac{F_1}{|B|} \sum_{(i,j) \in B} \mathcal{S}_{ij} \tag{10}$$

Here, $F_1$ reflects the alignment quality between the full sets of generated and ground-truth utterances:

$$\mathrm{Prec} = \frac{|B|}{|\hat{\mathcal{U}}|}, \quad \mathrm{Recall} = \frac{|B|}{|\mathcal{U}|}, \quad F_1 = \frac{2 \cdot \mathrm{Prec} \cdot \mathrm{Recall}}{\mathrm{Prec} + \mathrm{Recall}} \tag{11}$$

### D.3 TIMING ACCURACY SCORE

The timing accuracy score is defined as a weighted sum of three components: the **start score**, **end score**, and **overlap score**. We begin with the first two that are calculated over the matched pairs in $B$:

$$S_{ij}^{\mathrm{start}} = \exp\left(-\frac{|s_i - \hat{s}_j|}{\tau_{\mathrm{pen}}}\right), \quad S^{\mathrm{start}} = \frac{F_1}{|B|} \sum_{(i,j) \in B} S_{ij}^{\mathrm{start}} \tag{12}$$

$$S_{ij}^{\mathrm{end}} = \exp\left(-\frac{|e_i - \hat{e}_j|}{\tau_{\mathrm{pen}}}\right), \quad S^{\mathrm{end}} = \frac{F_1}{|B|} \sum_{(i,j) \in B} S_{ij}^{\mathrm{end}} \tag{13}$$

where $\tau_{\mathrm{pen}}$ is a scaling factor that controls how severely to penalize temporal misalignment.

To discourage overlapping utterances, which can result in fragmented or poorly-timed interactions, we define a penalty for each generated utterance $\hat{u}_j \in \hat{\mathcal{U}}$ based on its overlap with all other generated utterances:

$$\mathrm{overlap}(j, j') = \min(\hat{e}_j, \hat{e}_{j'}) - \max(\hat{s}_j, \hat{s}_{j'}), \quad O_j = \sum_{j \neq j'} \max(0, \mathrm{overlap}(j, j')) \tag{14}$$

$$S_j^{\mathrm{overlap}} = \exp\left(-\frac{O_j}{\tau_{\mathrm{pen}}}\right), \quad S^{\mathrm{overlap}} = \frac{F_1}{|\hat{\mathcal{U}}|} \sum_{j \in \hat{\mathcal{U}}} S_j^{\mathrm{overlap}} \tag{15}$$

Note that $\mathrm{overlap}(j, j')$ is positive only when utterances temporally intersect; otherwise, it is clamped to zero.

The total timing accuracy score is then:

$$S^t = \alpha_{\mathrm{start}} S^{\mathrm{start}} + \alpha_{\mathrm{end}} S^{\mathrm{end}} + (1 - \alpha_{\mathrm{start}} - \alpha_{\mathrm{end}}) S^{\mathrm{overlap}} \tag{16}$$

where $\alpha_{\mathrm{start}}$ and $\alpha_{\mathrm{end}}$ are tunable weights. We set $\tau_{\mathrm{pen}}$ to 1.0 for all our experiments.

## D.4 FINAL SCORE

The final **TRACE** score combines semantic and timing accuracy into a single metric:

$$\text{TRACE} = \alpha S^a + (1 - \alpha)S^t \tag{17}$$

where $\alpha$ is a tunable weight that balances the two components. For all evaluations, we use the following TRACE parameters:

$$\alpha_{\text{start}} = 0.4, \quad \alpha_{\text{end}} = 0.4, \quad \alpha = 0.5 \tag{18}$$

TRACE is designed to be decomposable, enabling detailed analysis of VLM performance in real-time settings. By jointly evaluating *what* is said and *when* it is said against gold-standard human utterances, TRACE not only reflects the dual requirements of real-time interaction—adapting to new observations (perceptual updating) and responding to the consequences of prior actions (contingency awareness)—but also helps ensure that generated utterances feel natural to humans. We hope that TGLG and TRACE provide a useful foundation for future research on automatic evaluation metrics in real-time, interactive settings.

## E ACTION CATEGORIES

- **Attempts**: Shots on target, Shots off target, Clearance
- **Discipline**: Yellow card, Red card, Yellow→red card
- **Goal/Penalty**: Goal, Penalty
- **Infractions**: Offside, Foul
- **Restarts**: Kick-off, Ball out of play, Throw-in, Corner, Direct free-kick, Indirect free-kick
- **Substitution**: Substitution

## F TASK CATEGORIES

- **Assemble Furniture**: assemble nightstand, assemble stool, assemble tray table, assemble utility cart
- **Disassemble Furniture**: disassemble nightstand, disassemble stool, disassemble tray table, disassemble utility cart
- **Make Coffee**: make coffee with nespresso machine, make coffee with espresso machine
- **Repair Machinery**: change belt, change circuit breaker, fix motorcycle
- **Setup Electronics**: setup camera, setup switch, setup big printer, setup small printer, setup gopro, assemble laser scanner, assemble computer

## G QUALITATIVE EXAMPLES

### G.1 PERCEPTUAL UPDATING

Below are qualitative examples for perceptual updating (SoccerNet). Proper-noun mismatch is expected, as the models were neither trained nor provided with the necessary context to handle proper nouns such as player names. Replacing proper nouns with NER tags like `<PLAYER>` did not change TRACE scores much, so we use the utterances with the original proper nouns.

### G.1.1 VIDEOLLM-ONLINE EXAMPLES

We did not find clear successful examples for VideoLLM-Online.

**Delayed Start**
**VIDEO / TIME:** `2015-08-29 - 17-00 Manchester City 2 - 0 Watford/2_224p`@1617.7s

**GENERATED:** "And on to Silva, and on to Aguero, and away by . . ."
**GROUND TRUTH:** "And here goes Sergio Aguero."
**SEMANTIC:** 0.826 **TIMING:** 0.011 **START:** 0.007 **END:** 0.000 **OVERLAP:** 0.041

*The utterance is semantically relevant but begins 5s late.*

**Premature Cutoff**
**VIDEO / TIME:** 2015-02-24 – 22-45 Manchester City 1 – 2 Barcelona/1_224p@2596.6s
**GENERATED:** "Here's Milner."
**GROUND TRUTH:** "Oh, it's a kind bounce off Milner now for Dani Alves."
**SEMANTIC:** 0.796 **TIMING:** 0.617 **START:** 0.993 **END:** 0.049 **OVERLAP:** 1.000

*Generation begins on time but ends 3s before the human commentary, omitting crucial detail.*

**Overlap**
**VIDEO / TIME:** 2015-09-19 – 19-30 Manchester City 1 – 2 West Ham/1_224p@205.4s
**GENERATED:** "And that's it, that's all you need, one little touch to get it away from the danger zone. . ."
**GROUND TRUTH:** "Collar off with a lovely ball for De Bruyne."
**SEMANTIC:** 0.717 **TIMING:** 0.003 **START:** 0.009 **END:** 0.000 **OVERLAP:** 0.000

*The model speaks for too long and drifts beyond the relevant action, producing an utterance that overlaps with subsequent events.*

### G.1.2 VLM-TSI EXAMPLES

**Success**
**VIDEO / TIME:** 2016-11-19 – 18-00 Manchester United 1 – 1 Arsenal/1_224p@2727.7s
**GENERATED:** "Sanchez."
**GROUND TRUTH:** "Sanchez again."
**SEMANTIC:** 0.916 **TIMING:** 0.962 **START:** 0.990 **END:** 0.914 **OVERLAP:** 1.000

**Delayed Start**
**VIDEO / TIME:** 2015-02-24 – 22-45 Manchester City 1 – 2 Barcelona/1_224p@1699.7s
**GENERATED:** "Iniesta."
**GROUND TRUTH:** "Iniesta."
**SEMANTIC:** 1.000 **TIMING:** 0.263 **START:** 0.077 **END:** 0.079 **OVERLAP:** 1.000

*VLM-TSI produces a semantically accurate utterance, but reacts a few seconds late to the event. The timing lag suggests room for faster perceptual updating.*

**Over-extended**
**VIDEO / TIME:** 2015-05-02 – 19-00 Atl. Madrid 0 – 0 Ath Bilbao/2_224p@2872.3s
**GENERATED:** "Another decent ball played in, but once again, the offside flag up."
**GROUND TRUTH:** "It's offside again."
**SEMANTIC:** 0.878 **TIMING:** 0.616 **START:** 0.983 **END:** 0.056 **OVERLAP:** 1.000

*VLM-TSI triggers at the correct moment but runs longer than the human commentator, overshooting the natural endpoint.*

**Premature Cutoff**
**VIDEO / TIME:** 2017-01-31 – 23-00 Liverpool 1 – 1 Chelsea/1_224p@1105.3s
**GENERATED:** "Here's Matic."

GROUND TRUTH: "Here's Matic, he's closed down quickly there by Roberto Firmino, didn't have the time that he thought he'd got."
SEMANTIC: 0.819   TIMING: 0.596   START: 0.981   END: 0.008   OVERLAP: 1.000

*VLM-TSI triggers at the correct moment but cuts off early, failing to capture the full play description present in the human commentary.*

### G.2   EXAMPLES OF IMPROVEMENTS OF VLM-TSI OVER VIDEOLLM-ONLINE

**Big Timing Gain**
VIDEO / TIME:          2016-01-03 – 16-30 Crystal Palace 0 – 3 Chelsea/2_224p@2337.0s
VLM-TSI GENERATED: "Here's Costa."
VIDEOLLM-ONLINE GENERATED: long, off-topic filler ("And it's. . . I mean, they've just. . . They've been. . . ")
GROUND TRUTH: "Costa."
VLM-TSI SEM: 0.904   T: 0.799   S: 0.780   E: 0.718   O: 1.000
VIDEOLLM-ONLINE SEM: 0.540   T: 0.023   S: 0.058   E: 0.000   O: 0.000

*VLM-TSI calls the play at the exact moment; VideoLLM-Online is late and drifts into rambling commentary.*

**Fixes Overlap**
VIDEO / TIME:          2015-09-19 – 19-30 Manchester City 1 – 2 West Ham/1_224p@2669.4s
VLM-TSI GENERATED: "Sergio Aguero."
VIDEOLLM-ONLINE GENERATED: "It's a big chance, . . . , and he's right."
GROUND TRUTH: "It's Aguero!"
VLM-TSI SEM: 0.859   T: 0.611   S: 0.619   E: 0.407   O: 1.000
VIDEOLLM-ONLINE SEM: 0.673   T: 0.021   S: 0.050   E: 0.000   O: 0.006

*VLM-TSI delivers a concise, non-overlapping call; VideoLLM-Online rambles and overlaps heavily.*

### G.3   CONTINGENCY AWARENESS

Below are qualitative examples for contingency awareness (HoloAssist).

### G.3.1   VIDEOLLM-ONLINE EXAMPLES

**Success**
VIDEO / TIME: z102-july-13-22-dslr@69.9s
GENERATED: "Now please turn on the camera."
GROUND TRUTH: "Please turn on the camera."
SEMANTIC: 0.952   TIMING: 0.842   START: 0.883   END: 0.722   OVERLAP: 1.000

**Delayed Start**
VIDEO / TIME: z127-aug-10-22-printer_big@23.1s
GENERATED: "You're almost there! Just place the paper stack. . . "
GROUND TRUTH: "The next step is to add some paper in the printer. . . "
SEMANTIC: 0.879   TIMING: 0.010   START: 0.018   END: 0.004   OVERLAP: 0.005

*The instruction is correct but arrives several seconds after the user has already loaded paper.*

**Delayed Start**
VIDEO / TIME: z198-sep-18-22-dslr@54.3s
GENERATED: "I'll repeat. We will now detach the lens cover."
GROUND TRUTH: "The next step is to detach the lens cover."
SEMANTIC: 0.846   TIMING: 0.016   START: 0.013   END: 0.005   OVERLAP: 0.045

*Again, the command is semantically correct but lags behind the moment the human has already started removing the cover.*

**Incorrect Correction**
**VIDEO / TIME:** `R194-25Oct-GraphicsCard`@54.3s
**GENERATED:** "That's it! You've successfully removed the graphics card."
**GROUND TRUTH:** "Outside. On the outside."
**SEMANTIC:** 0.497   **TIMING:** 0.599   **START:** 0.699   **END:** 0.299   **OVERLAP:** 1.000

*Despite good timing, the model declares task completion for the wrong object, demonstrating a causal misunderstanding of its own prior instruction.*

### G.3.2   VLM-TSI EXAMPLES

**Success**
**VIDEO / TIME:** `R124-15Aug-Circuit-Breaker`@119.1s
**GENERATED:** "Go ahead and press that button."
**GROUND TRUTH:** "Go and click that."
**SEM:** 0.752   **TIMING:** 0.751   **START:** 0.552   **END:** 0.824   **OVERLAP:** 1.000

**Delayed Start**
**VIDEO / TIME:** `z092-july-11-22-knarrevik_assemble`@23.1s
**GENERATED:** "You're doing well! You've inspected the table..."
**GROUND TRUTH:** "Please start by assembling one leg to one table."
**SEMANTIC:** 0.823   **TIMING:** 0.203   **START:** 0.007   **END:** 0.000   **OVERLAP:** 1.000

*The instruction arrives too late, failing to guide the user's immediate next step.*

**Incorrect Correction**
**VIDEO / TIME:** `z018-june-20-22-printer_big`@17.3s
**GENERATED:** "You're doing great! Remember to take your time..."
**GROUND TRUTH:** "It's the moon icon on the right."
**SEMANTIC:** 0.468   **TIMING:** 0.595   **START:** 0.956   **END:** 0.031   **OVERLAP:** 1.000

*Despite being well-timed, the guidance is irrelevant to the user's current task, showing a misunderstanding.*

### G.4   EXAMPLES OF IMPROVEMENTS OF VLM-TSI OVER VIDEOLLM-ONLINE

**Semantic & Timing Gain**
**VIDEO / TIME:** `R200-7Nov-ATV`@47.7s
**VLM-TSI GENERATED:** "That's the one! Now, adjust the shift arm on the mounting peg. Make sure it's securely attached."
**VIDEOLLM-ONLINE GENERATED:** "You're doing great! Keep going."
**GROUND TRUTH:** "Yeah, you want to attach that onto, mmmm the mounting peg."
**VLM-TSI   SEM:** 0.724   **T:** 0.874   **S:** 0.711   **E:** 0.974   **O:** 1.000
**VIDEOLLM-ONLINE   SEM:** 0.506   **T:** 0.559   **S:** 0.847   **E:** 0.049   **O:** 1.000

*VLM-TSI delivers the correct, specific instruction at the precise moment, while VideoLLM-Online issues a generic prompt without adapting to the user's actions.*