# OpenReview forum: "Temporally-Grounded Language Generation: A Benchmark for Real-Time Vision-Language Models"
_ICLR.cc/2026/Conference — ICLR 2026 Conference Withdrawn Submission_

### Official Review · Reviewer_f4Ef · 2025-10-20

**Soundness:** 2
**Presentation:** 1
**Contribution:** 1
**Rating:** 2
**Confidence:** 4

**Summary:**

The paper introduces Temporally-Grounded Language Generation (TGLG), a benchmark designed to evaluate the performance of real-time video language models (VLMs) that must produce time-synchronized responses while processing streaming video input. The benchmark builds on two existing datasets, SoccerNet (Cioppa et al., 2022), featuring sports broadcast videos, and HoloAssist (Wang et al., 2023), comprising egocentric human–object interaction videos. TGLG is organized around two complementary tasks: perceptual updating, which measures how proactively a model responds to evolving visual input, and contingency awareness, which assesses how effectively it provides timely, context-appropriate feedback. The authors also propose TRACE, a metric that jointly evaluates semantic similarity and temporal alignment between predicted and ground-truth utterances. The benchmark primarily evaluates VideoLLM-Online (Chen et al., CVPR 2024) and a modified variant that omits the end-of-sentence (EOS) token to avoid silence gaps. Experimental results show modest but consistent improvements with this modification.

**Strengths:**

- The paper targets an important and timely problem of evaluating real-time multimodal reasoning in VLMs. As such, it contributes to an emerging and rapidly evolving research direction.
- The use of two complementary datasets with distinct characteristics (third-person sports vs. first-person human interaction) is a thoughtful design choice that enhances the benchmark’s generality. The inclusion of cross-dataset evaluation (HoloAssist is used just for testing) further strengthens its robustness.
- The benchmark provides fine-grained task/action groupings, allowing analysis of model behavior under varying temporal and semantic demands. This granularity can help identify where current systems struggle with temporal grounding.

**Weaknesses:**

- As a benchmark paper, the experimental evaluation is rather limited. The authors only assess two versions of VideoLLM-Online, omitting several strong recent baselines such as Stream-VLM (Panchal et al., 2024), FlashVStream (Zhang et al., 2024), Dispider (Qian et al., 2025), StreamChat (Xiong et al., 2025), and StreamChat (Liu et al., 2025). Including or at least discussing results from these models would significantly strengthen the empirical validation.
- The related work section does not sufficiently situate TGLG within the landscape of contemporary benchmarks such as QEVD (Panchal et al., 2024), OmniMMI (Wang et al., CVPR 2025), and OVO-Bench (Niu et al., 2025). A detailed comparison would clarify the unique contribution and scope of TGLG.
- The proposed model modification (ignoring the EOS token) is minimal and should not be framed as a novel modeling contribution.
- The TRACE metric introduces several empirically tuned hyperparameters, which raises concerns about reproducibility and interpretability. Furthermore, it is unclear whether a joint score is preferable to reporting separate measures of semantic and temporal alignment (see related question below).
- The presentation could be improved, especially in sections explaining motivation and task setup (e.g., lines 139–147 and 256–269, and Figure 1). Given the multimodal nature of the work, clearer visual-textual illustrations would aid reader understanding. Moreover, the related work section omit a detailed discussion of most related models and benchmarks.

Minor Issues:
- Watch, Talk and Guide (Bao et al., 2023) appeared in Findings of EMNLP 2023.
- StreamBench (Xiong et al., 2025) appeared in ICLR 2025.


**Missing References**
- Liu et al. StreamChat: Chatting with Streaming Video. ArXiv Preprint arXiv:2412.08646v, March 2025.
- Niu et al. OVO-Bench: How Far is Your Video-LLMs from Real-World Online Video Understanding? CVPR 2025
- Panchal et al. What to Say and When to Say it: Live Fitness Coaching as a Testbed for Situated Interaction. NeurIPS 2024 Track on Datasets and Benchmarks.
- Qian et al. Dispider: Enabling video LLMs with active real-time interaction via disentangled perception, decision, and reaction. CVPR 2025.
- Wang et al. OmniMMI: A Comprehensive Multi-modal Interaction Benchmark in Streaming Video Contexts. CVPR 2025.

**Questions:**

1. The benchmark currently evaluates two architecturally identical VLMs. How might differences in inference latency or token generation rate across models affect real-time responsiveness and TRACE scores?
2. The TRACE metric combines semantic and temporal components via a weighted sum. Why not report these two aspects independently, as done in (Panchal et al., 2024)?
3. To strengthen the empirical evaluation, please consider including additional recent real-time VLMs or at least discussing their performance relative to TGLG’s objectives. Similarly, a comparative discussion with existing streaming benchmarks (see Weaknesses) would help clarify the distinct contributions of this work.

---

> ### Author Response · Authors · 2025-11-20
>
> Thank you for your detailed review and for highlighting the importance of evaluating multimodal models under real-time streaming conditions. We are glad to hear that you find our benchmark robust through its use of two complementary datasets, as well as the modular design of our metric that allows for fine-grained analyses of model performance. We address each of your concerns and questions below.
>
> **Weaknesses**
>
> 1. Limited scope of evaluation and benchmark selection
>
> Thank you for suggesting additional streaming VLMs and benchmarks such as Stream-VLM, FlashVStream, Dispider, StreamChat (Xiong et al., 2025), StreamChat (Liu et al., 2025), QEVD, OmniMMI, and OVO-Bench. We acknowledge their contributions. However, all of these systems rely on turn-based or point-event assumptions--text input and responses are treated as single events while the environment is paused. This aligns with VideoLLM-Online, which we already evaluate and surpass, and does not model the real-time visual and linguistic streams that TGLG is designed to test.
>
> StreamChat (Liu et al., 2025) is closest to our setting, as it recognizes separate visual and textual streams. However, it still makes the turn-based assumption that “the instruction is input instantaneously” and uses LLM-generated conversations with timestamps rather than naturalistic human dialogue. Its architecture also employs cross-attention mechanisms, whereas VLM-TSI shows that simple input-level interleaving already enables real-time grounding. We will include StreamChat as a baseline once code and training data are publicly released.
>
> QEVD (Panchal et al., 2024) is the most relevant benchmark among those listed. However, its interactions are limited: utterances are short commands, and communication is one-directional (instructor -> student). To evaluate contingency awareness under natural two-sided interaction, we selected HoloAssist, which contains richer human dynamics and better tests this capability.
>
> In summary, the suggested models and benchmarks do not support evaluation of real-time perceptual updating and contingency awareness under continuously interleaved visual-textual input. We will include explicit discussion of these works in Section 2 of the next version.
>
> 2. Minimal model modification
>
> The primary modification in VLM-TSI is the time-synchronized token interleaving strategy described in Section 5.1. A secondary consequence of this design is that there is no need to tune the EOS threshold used in VideoLLM-Online--instead, VLM-TSI naturally uses the BOS token to signal when to speak (Section 5.2). Therefore, the assertion that “ignoring the EOS token” is the only modification is not accurate, as it follows from the broader input-level synchronization strategy that defines VLM-TSI. We will add a brief clarification in Section 5 to better highlight that EOS handling is a consequence of the interleaving strategy, not its primary mechanism.
>
> 3. Concerns about design choices of TRACE
>
> TRACE is modular, and all sub-metrics (semantic accuracy, start score, end score, overlap score, and F1) are reported independently in our evaluation tables. The joint score uses nearly equal weights and was not heavily tuned; in fact, the overlap score is given a lower weight (0.2 vs. 0.4 for start/end; Section D.4, Eq. 18), which makes the estimate of VLM-TSI’s gains conservative, as it performs best on overlap.
>
> This approach follows widely used composite metrics such as F1, BLEU, CIDEr, and METEOR, which also use manual weighting and averaging strategies to produce a concise summary score. Such metrics enable rapid comparison and iteration while preserving access to fine-grained components. We will add a brief note in Section 4.3 to clarify this rationale.
>
> 4. Concerns about presentation and related work
>
> Thank you for the feedback on presentation and related work. We appreciate that you found the problem “important and timely” and the framework “thoughtful,” “robust,” and providing granular analyses--points echoed by other reviewers as well. We agree there is room for further clarity, and will refine the figures and update Section 2 to incorporate the contemporary models and benchmarks you mentioned in the next version.
>
> **Questions**
>
> 1. Inference speed’s effect on responsiveness and TRACE
>
> This question relates directly to the core assumption that TGLG and TRACE are designed to evaluate. If a VLM’s decoding latency was zero, it would essentially be making the turn-based/point-event assumption (L46–48, L103–106, L299–300), causing overlaps and delays. TRACE is designed to measure these defects. Please see our response to Reviewer sNsp (weakeness 1) for more detail.

---

> > ### Comment · Reviewer_f4Ef · 2025-11-26
> >
> > Thank you for the detailed response. While I appreciate the clarifications and the promised improvements to presentation and related work, I do not feel that the rebuttal adequately addresses my primary concern that **the empirical evaluation is limited in scope**.
> >
> > Many of the suggested baselines (including StreamChat, FlashVStream, Dispider) can operate on continuous streams or can be reasonably adapted to the streaming setting. Dismissing them as "turn-based" does not fully justify omitting them, especially for a benchmark submission where **representative and diverse baselines are essential**. Similarly, the explanation regarding TRACE and latency effects remains high-level and does not clarify model-dependent differences.
> >
> > In addition, **the authors did not upload a revised version of the paper**, even though ICLR allows revised manuscripts during the rebuttal. This is important, as several of the issues I raised (e.g., missing related work, unclear illustrations, insufficiently described assumptions) would be better evaluated with updated text, figures, or tables rather than narrative promises. The absence of a revised version makes it difficult to assess the extent to which the authors can actually improve the clarity and positioning of the paper.
> >
> > Overall, although the benchmark formulation is interesting, I still believe the submission would benefit from a more comprehensive empirical study and a more detailed comparative positioning against closely related benchmarks.

---

> > > ### Author Response · Authors · 2025-11-29
> > >
> > > Thank you for the follow-up and for clarifying your concerns. We agree that additional baselines would further strengthen future iterations, and we will incorporate the contemporary works you mentioned in the next version. For this submission, we focused on establishing the real-time problem formulation and therefore selected VideoLLM-Online as the primary baseline, as its assumptions are the closest to the real-time setting we propose among contemporary models at the time of writing. Expanding the empirical scope is a natural next step, and we appreciate your suggestions in guiding this direction.
> > >
> > > Thank you again for the constructive engagement.

---

### Official Review · Reviewer_sNsp · 2025-11-01

**Soundness:** 3
**Presentation:** 3
**Contribution:** 2
**Rating:** 6
**Confidence:** 4

**Summary:**

This paper introduces a new task and benchmark for Temporally-Grounded Language Generation (TGLG), which aims to incorporate two core capabilities of perceptual updating and contingency awareness into real-time interactive video understanding settings. Specifically, this work builds two subsets based on real-time soccer game commentary and egocentric human interaction videos, and a new temporally synchronized token interleaving strategy is proposed to tackle the new challenges. Quantitative experiments are conducted to show the effectiveness of the proposed method and the benchmark.

**Strengths:**

1.The motivation of incorporating the capabilities of perceptual updating and contingency awareness into real-time video-llms is practical and intuitively reasonable.

2.The proposed benchmark, metric and method in this work are shown to be effective under real-time settings.

**Weaknesses:**

1.As the authors mentioned in the manuscript, the existing turn-based video-llms would give response to the environment with overly high latency, which is a major obstacle for them to handle the real-time settings. However, these video-llms are generally at a large size and have huge amout of parameters which naturally make them unsuitable for real-time response. What if the turn-based video-llms are optimized to have fewer parameters and faster response speed, for example, turn-based models could also generate response promptly if they can finish decoding before new frames come in. Are there any discussions or experiments to analyze this aspect?

2.For the proposed temporally synchronized vision and text token interleaving strategy, it is shown to be effective to generate real-time response conditioned on fastly evolving visual environments. But based on my understanding, such highly fragmented token mixing method would inevitably destroy the coherence of the visual input and also the textual context, especially when the full off-line inputs are available. So are there any experiments or analysis on the performance of such strategy under a off-line setting? Will it largely decrease the performance of the model under off-line settings?

**Questions:**

Please refer to the weaknesses.

**Details Of Ethics Concerns:**

N/A.

---

> ### Author Response · Authors · 2025-11-20
>
> Thank you for your valuable feedback. We appreciate that you found our work has clear motivations for addressing key capabilities in real-time settings. We're also glad that you found our proposed benchmark, metric, and baseline model effective in this context. We address your concerns below.
>
> **Weaknesses**
>
> 1. Turn-based VLMs with a fast response time
>
> We appreciate your thoughtful question about whether a turn-based VLM with fast decoding speed could be sufficient for real-time settings. This is in fact a scenario that TGLG and TRACE are specifically designed to evaluate, and one that VLM-TSI seeks to address.
>
> Consider an extreme version of this scenario: a turn-based VLM that can generate a response instantaneously (i.e., 0-second latency). This is essentially the assumption made by existing models like VideoLLM-Online, which treat text generation as a point event (L46–48, L103–106, L299–300). Another way to describe this assumption is that the environment “pauses” while the model generates a response. In such cases, delays due to slow decoding would not occur. However, in real-time settings where text tokens are spoken or streamed to users, the outputs of such models can temporally overlap--responses are triggered immediately upon new events, without regard for whether the previous utterance has finished. This leads to overlapping speech, confusing output in subtitle-style interfaces, or rapid flashes of text that are difficult to follow. While responses could technically be queued and displayed sequentially, this breaks down in interactive contexts where timing and situational relevance are critical.
>
> A carefully designed UI system could potentially manage these overlapping outputs—for example, by merging or suppressing them--but doing so would require external logic to track environmental changes and revise the displayed or spoken output accordingly. In effect, such a system would need to perform perceptual updating and contingency-aware revision, but outside the model itself. This is precisely the type of capability we aim to evaluate within the model, and which motivates our proposed benchmark and TRACE metric.
>
> To this end, TRACE includes an overlap score (S^{overlap}) that explicitly penalizes responses that temporally overlap. Our experimental results (Section 6.2.1) show that VLM-TSI significantly reduces this issue by synchronizing inputs and outputs in real time to avoid misalignment.
>
> 2. VLM-TSI’s performance in offline settings
>
> We appreciate your question about VLM-TSI’s suitability for offline settings. VLM-TSI is specifically designed for real-time settings, where inputs arrive continuously and responses must adapt incrementally. Our benchmark, model, and evaluation methodology are all scoped to this real-time context.
>
> As you noted, the token interleaving strategy used in VLM-TSI results in input distributions that differ from those used in conventional offline tasks. This mismatch may adversely affect the model’s performance in offline settings. That said, since the underlying LLM is kept frozen and only the projection layer mapping visual features into the LLM’s embedding space is trained, it is possible that the LLM’s strong generation prior could mitigate some of this mismatch.
>
> Evaluating performance in offline settings could be an interesting direction for future work, particularly for understanding how different interleaving strategies interact with frozen language models across deployment scenarios.

---

### Official Review · Reviewer_TqVq · 2025-11-01

**Soundness:** 2
**Presentation:** 2
**Contribution:** 3
**Rating:** 2
**Confidence:** 3

**Summary:**

This paper introduces a new task for testing vision-language models in online real-time settings, which is called as Temporally-Grounded Language Generation (TGLG). The proposed benchmark includes sports broadcasting and egocentric videos. Within this work, TRACE, a novel evaluation metric, is also introduced to evaluate models on the TGLG benchmark. Furthermore, the authors introduce VLM-TSI, which enables processing interleaved vision and language tokens in a time-synchronized way, later tested on the repurposed benchmark using the proposed metric TRACE. The downstream task experiments reveal that the further finetuning the baseline using VLM-TSI approach on SoccerNet and HoloAssist datasets improve real-time spatio-temporal processing based on the proposed metric.

**Strengths:**

- Research on an interesting direction even within the domain of spatio-temporal vision-language learning: time-synchronized processing in real-time scenarios.
- Substantial improvements over the baseline.

**Weaknesses:**

- No human validation study on how the proposed metric aligns with human preferences.
- I'm confused about this work's purpose. It seems that the data resources already exist, and VLM-TSI method actually proposes finetuning a suitable model on these resources. According to my current understanding, the task is actually not novel, the methodology is actually to finetune a proper model on time-synchronized interleaved video-language data. The only actual novelty is the proposed metric, which is not evaluated.
- Dataset-specific finetuning: It appears that this work performs two separate finetuning on two separate downstream datasets. It would be good to expand the experiments considering both data resources simultaneously.
- Limited evaluation: There is only one baseline in the current evaluation setup. I am aware the fact that the other models would not be good baselines as suggested by authors. However, expanding evaluations to more general zero-shot video-language benchmarks (e.g., Video-MME) could be beneficial. For instance, did the model become better in spatio-temporal processing after learning more real-time dynamics?

**Questions:**

- L428-L431: Could this be more related to models finding shortcuts through patterns? Penalty position could be detected by using single frame, no spatio-temporal processing is required actually.
- Fig. 1, and Fig. 4 could be combined to explain the task in a better way. Same goes for Fig 2. and Fig 3., they do not need to be separate.

---

> ### Author Response · Authors · 2025-11-20
>
> Thank you for your review and for noting the novel direction of our research, as well as the substantial improvements our proposed baseline model makes over prior works. We address your concerns and questions below.
>
> **Weaknesses**
>
> 1. On the need for human evaluation
>
> TRACE is designed to evaluate two capabilities central to real-time settings: perceptual updating and contingency awareness, corresponding to objective properties--semantic accuracy conditioned on visual state and precise temporal alignment between utterances and events. Since these can be measured directly using ground-truth timestamps and content, human evaluation is not required for TRACE’s intended purpose.
>
> Human judgments may assess fluency or conversational style, but they do not provide additional insight into temporal correctness or event alignment. Automatic evaluation ensures consistent, fast, and fair comparison across models--a key requirement for advancing real-time systems. For these reasons, human evaluation is not the most appropriate tool for the capabilities that TGLG and TRACE aim to measure.
>
> 2. Clarifying the purpose of TGLG
>
> You raise an important point about the underlying data sources. While SoccerNet and HoloAssist exist, they have never been used to evaluate real-time perceptual updating or contingency awareness, nor do the original evaluation protocols measure these capabilities. SoccerNet’s evaluation tasks--for example dense video captioning, ball action spotting, and multi-view foul recognition--make turn-based assumptions and focus on offline understanding of soccer matches. HoloAssist’s evaluation tasks--such as mistake detection and intervention type prediction--similarly treat predictions as point events under a turn-based setup. Neither benchmark captures whether a model produces timely, continuously updated utterances that remain aligned with an evolving visual stream.
>
> TGLG introduces a new task definition, new evaluation protocol, and a new metric that jointly accounts for semantic accuracy and temporal alignment in order to measure real-time behavior. The contribution lies in reframing existing raw data into a benchmark that can meaningfully probe capabilities that prior evaluations overlook. We will clarify this distinction more explicitly in the next version.
>
> 3. Training jointly on both data sources
>
> Training on both datasets jointly is certainly possible and would be appropriate for developing a more comprehensive model. However, our goal in this work is to isolate two distinct capabilities:
>
> - SoccerNet primarily probes perceptual updating under rapid, high-frequency changes.
> - HoloAssist probes contingency awareness in egocentric, action-dependent guidance.
>
> Training on each dataset independently allows us to attribute model behavior to the specific capability being evaluated, without confounding effects introduced by multi-domain optimization. We will clarify this design choice in the next version.
>
> 4. Evaluation on additional zero-shot video-language benchmarks
>
> We appreciate the suggestion to evaluate on broader zero-shot benchmarks such as Video-MME. However, these benchmarks assume turn-based generation and offline full-context access--the model sees the entire video before producing a single answer. Applying TGLG-style real-time requirements would require substantial changes to annotation format, temporal structure, and scoring logic--effectively altering their intended purpose. Even if modified to use only past frames (e.g., dense captioning), this still reflects the point-event assumption used by VideoLLM-Online, where generation does not remain aligned with an evolving visual stream. One could also feed an entire video to VLM-TSI and narrate it in real time before extracting an answer post-hoc, but this introduces confounding factors unrelated to temporal grounding and would require a multi-stage pipeline. We view this as orthogonal to TGLG’s goals.
> Evaluating VLM-TSI on offline zero-shot benchmarks is a promising direction, but lies outside the present scope, which focuses on studying key real-time capabilities--perceptual updating and contingency awareness--in a controlled and principled manner.
>
> **Questions**
>
> 1. On “shortcut” explanations for penalty events
>
> Your intuition aligns with our own hypothesis. As noted in the paper (L428–431), “Goal/Penalty” events are temporally atomic and typically covered with a single utterance, leading to a smaller gap between VLM-TSI and VideoLLM-Online. These events are also often recognizable from a single or very short sequence of frames, reducing the need for extended temporal integration. We agree both factors likely contribute to the reduced disparity and will clarify this in the next version.
>
> 2. Combining figures
>
> Thank you for the helpful suggestion. We will adjust the figures accordingly to make the paper easier to understand.

---

> > ### Comment · Reviewer_TqVq · 2025-11-27
> >
> > I have read the authors' response and decided to maintain my rating.
> >
> > > Human judgments may assess fluency or conversational style, but they do not provide additional insight into temporal correctness or event alignment.
> >
> > So, do you mean that humans don't have the ability to comprehend about spatiotemporal events?

---

> > > ### Author Response · Authors · 2025-11-29
> > >
> > > Thank you for your follow-up. We did not intend to imply that humans lack spatiotemporal understanding; rather, our point was that human evaluation would likely rely on the same objective criteria that TRACE already measures, such as temporal alignment between events and utterances. In that sense, human evaluation would not necessarily add new information beyond what TRACE already captures in a more systematic and reproducible way.
> > >
> > > Importantly, TRACE is grounded in human behavior: the reference utterances come from real human commentary and conversations, so high TRACE scores, especially timing accuracy, indicate greater alignment with how humans naturally speak and respond in real time. We will clarify this in the final version to avoid any confusion.

---

### Note · Authors · 2025-12-31

I have read and agree with the venue's withdrawal policy on behalf of myself and my co-authors.